# Transcription factor regulation of eQTL activity across individuals and tissues

Elise D. Flynn[1,2], Athena L. Tsu[2,3], Silva Kasela[1,2], Sarah Kim-Hellmuth[1,2,4], Francois Aguet[5], Kristin G. Ardlie[5], Harmen J. Bussemaker[1,6], Pejman Mohammadi[7,8]*, Tuuli Lappalainen[1,2,9]*

1 Department of Systems Biology, Columbia University, New York, New York, United States of America, 2 New York Genome Center, New York, New York, United States of America, 3 Department of Biomedical Engineering, Columbia University, New York, New York, United States of America, 4 Department of Pediatrics, Dr. von Hauner Children's Hospital, University Hospital, LMU Munich, Munich, Germany, 5 The Broad Institute of MIT and Harvard, Cambridge, Massachusetts, United States of America, 6 Department of Biological Sciences, Columbia University, New York, New York, United States of America, 7 Department of Integrative Structural and Computational Biology, The Scripps Research Institute, La Jolla, California, United States of America, 8 Scripps Translational Science Institute, The Scripps Research Institute, La Jolla, California, United States of America, 9 KTH Royal Institute of Technology, Stockholm, Sweden

* pejman@scripps.edu (PM); tlappalainen@nygenome.org (TL)

**Data Availability Statement:** Data generated by this study can be found in the Supplemental Tables or at https://github.com/LappalainenLab/TF-eQTL.

**Funding:** EDF was supported by NIH/NHGRI grant 5F31HG010580. EDF, ALT, HJB, and TL were

## Abstract

Tens of thousands of genetic variants associated with gene expression (*cis*-eQTLs) have been discovered in the human population. These eQTLs are active in various tissues and contexts, but the molecular mechanisms of eQTL variability are poorly understood, hindering our understanding of genetic regulation across biological contexts. Since many eQTLs are believed to act by altering transcription factor (TF) binding affinity, we hypothesized that analyzing eQTL effect size as a function of TF level may allow discovery of mechanisms of eQTL variability. Using GTEx Consortium eQTL data from 49 tissues, we analyzed the interaction between eQTL effect size and TF level across tissues and across individuals within specific tissues and generated a list of 10,098 TF-eQTL interactions across 2,136 genes that are supported by at least two lines of evidence. These TF-eQTLs were enriched for various TF binding measures, supporting with orthogonal evidence that these eQTLs are regulated by the implicated TFs. We also found that our TF-eQTLs tend to overlap genes with gene-by-environment regulatory effects and to colocalize with GWAS loci, implying that our approach can help to elucidate mechanisms of context-specificity and trait associations. Finally, we highlight an interesting example of IKZF1 TF regulation of an *APBB1IP* gene eQTL that colocalizes with a GWAS signal for blood cell traits. Together, our findings provide candidate TF mechanisms for a large number of eQTLs and offer a generalizable approach for researchers to discover TF regulators of genetic variant effects in additional QTL datasets.

## Author summary

Gene expression is regulated by local genomic sequence and can be affected by genetic variants. In the human population, tens of thousands of *cis*-regulatory variants have been

supported by NIH/NIMH grant R01MH106842.
EDF, HJB, and TL were supported by the Vagelos
Precision Medicine Pilot Grant from Columbia
University (https://precisionmedicine.columbia.
edu/Vagelos_Precision_Medicine_Pilot_Grant). SK
and TL were supported by NIH/NHLBI grant
R01HL142028. SKH was supported by Marie-
Sklodowska Curie fellowship H2020 grant 706636
and Reinhard-Frank Stiftung and the Helmholtz
Young Investigator grant VH-NG-1620. FA and
KGA were supported by NIH/NHLBI contract
HHSN268201000029C and NIH/NHGRI grant
5U41HG009494. HJB was supported by NIH/
NHGRI grant R01HG003008. PM was supported
by NIH/NIGMS grant R01GM140287 and NIH/
NCATS grant UL1TR002550. The funders had no
role in study design, data collection and analysis,
decision to publish, or preparation of the
manuscript.

**Competing interests:** I have read the journal's
policy and the authors of this manuscript have the
following competing interests: EDF is currently
employed by Patch Biosciences. TL advises Variant
Bio, Goldfinch Bio, GlaxoSmithKline and has equity
in Variant Bio. FA is an inventor on a patent
application related to TensorQTL.

discovered that are associated with altered gene expression across tissues, cell types, or
environmental conditions. Understanding the molecular mechanisms of how these small
changes in the genome sequence affect genome function would offer insight to the genetic
regulatory code and how gene expression is controlled across tissues and environments.
Current research efforts suggest that many regulatory variants' effects on gene expression
are mediated by them altering the binding of transcription factors, which are proteins that
bind to DNA to regulate gene expression. Here, we exploit the natural variation of TF
activity among 49 tissues and between 838 individuals to elucidate which TFs regulate
which regulatory variants. We find 10,098 TF-eQTL interactions across 2,136 genes that
are supported by at least two lines of evidence. We validate these interactions using func-
tional genomic and experimental approaches, and we find indication that they may pin-
point mechanisms of environment-specific genetic regulatory effects and genetic variants
associated to diseases and traits.

## Introduction

Gene expression is regulated by local genomic sequence and can be affected by genetic vari-
ants. In the human population, tens of thousands of *cis*-regulatory variants have been discov-
ered by expression quantitative trait locus (eQTL) mapping that associates genetic variation to
gene expression levels. These variants are enriched to fall in *cis*-regulatory elements and tran-
scription factor binding sites [1–3], implying that many eQTLs act via allelic difference in tran-
scription factor affinity. However, specific mechanisms of individual eQTL effects and their
variation across tissues or other contexts remain elusive. Understanding eQTL mechanisms, as
well as the contexts in which they are active, can shed light on the regulatory code of the
genome and how genetic variation perturbs this regulation.

Multiple efforts have sought to catalog eQTL effects across different contexts. The GTEx
Consortium profiled gene expression in 49 tissues across 838 donors and discovered eQTLs
for 1,260–18,795 genes per tissue [4,5]. Approximately a third of these eQTLs were estimated
to be active in all or almost all tissues, while a fifth were estimated to be active in five or fewer
tissues. Further work using computational cell type deconvolution has discovered approxi-
mately three thousand GTEx eQTLs whose effects are likely cell-type-specific [6]. Additional
context-specific eQTL effects have been assayed in a variety of settings, including during
immune stimulation [7,8], cell stress [9,10], cell differentiation [11], and drug or nutrient
exposure [12–14].

However, few studies have been conducted to investigate what causes eQTL context specific-
ity, i.e., why eQTLs are differentially active across contexts. Some of this variation is of course
explained by gene expression: genes that are not expressed will not have a measurable eQTL.
However, multiple studies have found that the link between gene expression and eQTL effect is
not straightforward, observing both increasing and decreasing allelic effects with increasing
gene expression [5,15]. When investigating the tissue variability of GTEx eQTLs, we discovered
that ~4% of eQTLs show increasing effects with increasing gene expression across tissues, and
~4% show decreasing effects [5]. These findings show that the context variability of eQTL effects
cannot be explained by gene expression alone and must depend on other features, such as chro-
matin accessibility, enhancer looping, or variable levels of transcription factor binding.

Determining eQTLs' mechanisms of action is challenging. The first obstacle lies in identify-
ing the causal variant(s) of a locus from the typically numerous associated variants in high
linkage disequilibrium (LD). Putatively functional variants can be pinpointed by statistical

fine-mapping approaches, complemented with genomic annotations such as regions of open chromatin, TF binding sites predicted by motifs, or allele-specific binding of TF ChIP-seq data [5,16–19]. However, these annotations suffer from both low specificity and low sensitivity. In terms of specificity, a large percentage of variants in the genome overlap some functional annotation; for instance, Gaffney et al found that 40% of SNPs in eQTLs overlapped a DNAse I hypersensitive site or histone-modified region [2]. In terms of sensitivity, functional data may be missing for the context in which the eQTL is active, and especially the highly informative allelic binding data are relatively sparse [20–22]. While experimental assays that directly measure regulatory effects of variants are increasing in scale, they may miss *in vivo* interactions or chromatin-specific regulation [23], and intensive experimental approaches to directly profile the effects and mechanisms of genetic variants in an eQTL [24–27] are difficult to conduct in a high-throughput manner.

One thing made clear by functional annotation data is that both eQTLs and chromatin-QTLs are enriched in known TF binding sites [2,4,28]. Given that TFs are one of the few sequence-specific interactors with the genome, it follows that noncoding eQTLs may exert their effects by altering TF binding, which would then affect chromatin accessibility, histone modifications, and gene expression. Adding to the hypothesis that TF binding may control eQTL variability, many cross-tissue eQTLs are enriched in TF binding sites for TFs with broad activity, while tissue-specific eQTLs are enriched for those relevant to their observed tissue [29]. By determining which TF's binding is being altered by an eQTL, we would be able to identify its mechanism of action, as well as understand what could be regulating the eQTL's context variability.

In this study, we set out to discover TF regulators of eQTLs by identifying eQTL effects that correlate with TF levels across or within tissues, using primarily GTEx data. We use the natural variation of TF levels between tissues, individuals, and conditions to elucidate mechanisms of action of eQTL regulatory variants and understand the context specificity of eQTL effects. We hypothesize that a portion of the observed context variability of an eQTL may be explained by the level of the TFs that bind to the eQTL to regulate gene expression (Fig 1A-1C). In the simplest form of the model, an allele may increase the affinity of an activating TF in a *cis*-regulatory site, which would lead to higher gene expression of that allele (Fig 1A). However, at low TF levels, the TF would not bind to either allele, resulting in the same low level of background gene expression from each allele. Conversely, at very high TF levels of saturated binding, even the lower affinity allele could bind the TF, and both alleles would have equal gene expression. This would translate to increasing and then decreasing eQTL effects as TF levels increase (Fig 1C). Other models are explored in S1 Fig.

Our approach links variation in TF levels to variation in eQTL effect size and requires no additional datatypes to be captured, using the same genetic and gene expression data that are used for eQTL discovery. It offers a novel approach to understanding regulatory variant context specificity that can refine and complement existing approaches based on statistical fine-mapping and functional genomic experiments. Applying it to GTEx data, we find thousands of interactions between TF levels and eQTL effects both across tissues and within tissues which represent potential TF regulators of eQTL effects, and we validate these data using numerous approaches and datasets. Finally, we highlight an example of an IKZF1-regulated eQTL that colocalizes with multiple GWAS blood traits, evidencing how this TF-based model can be used to unravel effects on human health and disease.

## Results

### Selection of putative regulatory variants

For the bulk of our analysis, we used the GTEx v8 dataset, including whole genome sequencing for 838 individuals and RNA sequencing from 73–706 samples across 49 tissues (S1 Table).

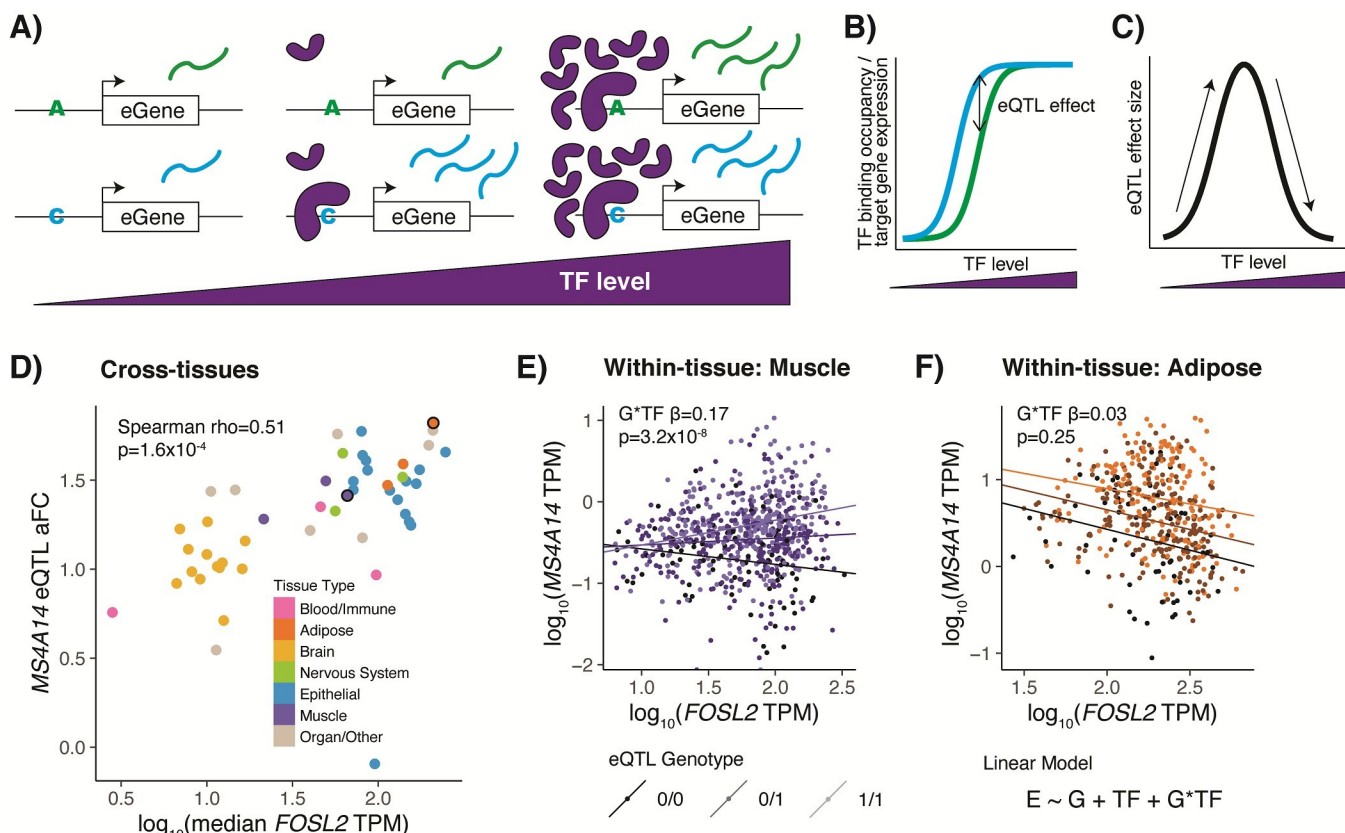

**Fig 1. TF model of eQTL effects. A)** TF binding to an eQTL variant with different allelic TF affinities is depicted at low, medium, and high TF levels. **B)** TF binding occupancy, resulting in target gene expression, for the two eQTL alleles across TF levels. **C)** Difference in expression of alleles or eQTL effect size, quantified as $\log_2$ allelic fold change, across TF levels. Our applied models only examine monotonic effects, which can be imagined as different sides of the hill. **D)** Tissues are plotted by eQTL effect size vs. median TF expression for an example MS4A14 eQTL and the FOSL2 TF. Cross-tissue TF-eQTL interactions are discovered by a Spearman correlation of these two measures, or with TF protein levels for the protein-based analysis. Two tissues circled in black are highlighted in the following panels. **E)** & **F)** Individuals are plotted by eGene expression vs. TF expression in Skeletal Muscle (E) or Adipose Visceral (F) tissue and are shaded by the genotype of the eQTL variant. Within-tissue TF-eQTL interactions are discovered using a multiple linear regression interaction model of normalized eGene expression by TF level, genotype, and TF level by genotype. Linear regression lines are plotted separately for each genotype, with corresponding G*TF interaction beta and p-value displayed on the chart. In Muscle, an eQTL is present, observable as a difference between the genotypes, and the difference gets larger as TF expression is higher, suggesting an interaction between TF level and eQTL effect. In Adipose, an eQTL is present, observable as a difference between the genotypes, but that difference does not appear to correlate with TF level.

We focused our analysis on common variants (>5% MAF) that have prior evidence of affecting gene expression and being regulated by a TF (Table 1). We used Caviar fine-mapping of GTEx eQTLs in 49 tissues to select variants that fell into a 95% credible set in at least one tissue [5,30]. We also required evidence that a TF binds in the vicinity of the variant. We focused our analysis on 169 TFs with both ENCODE ChIP-seq and HOCOMOCO motif information and included variants that overlapped at least one ChIP-seq peak and matched at least one motif for these TFs.

Filtering based on an intersection of these fine-mapping and functional annotations left us with 473,057 variants corresponding to 1,032,124 eQTLs across 32,151 genes. We found that 20% of lead tissue eQTL variants and 93% of all eGenes were included in the filtered variant set (S2 Fig). Each variant was associated with a median of two genes, and each gene was associated with a median of 28 variants across tissues (S3 Fig). Next, we used cross-individual and

**Table 1. GTEx variant annotations.**

| Dataset | >5% MAF GTEx variants | |
|---|---|---|
| | Count | Percent |
| All | 6,539,590 | - |
| Caviar fine-mapped set | 2,867,556 | 44% |
| ENCODE TF ChIP-seq peak | 1,425,613 | 22% |
| HOCOMOCO TF motif | 3,716,312 | 57% |
| Intersection | 473,057 | 7% |

Overlap of variants with >5% minor allele frequency (MAF) in the GTEx dataset that overlap various eQTL and TF annotations. Percent is based on all 5% MAF variants. Filtering eQTL variants for TF binding sites based on TF ChIP-seq peak overlaps and TF motif matches still results in a large number of potentially causal eQTL variants.

cross-tissue analyses to discover which of this large number of candidate variants had additional evidence of TF mechanisms underlying their eQTL effects.

## Interaction of eQTL effects and TF expression levels within tissues

We first investigated how inter-individual variation in TF levels within a tissue impacts eQTL effect size, with the hypothesis that such effects could represent TF regulators of specific eQTLs. We chose 20 diverse tissues that best represented all 49 GTEx eQTL tissues based on gene expression clustering (S4 Fig). For each of those tissues and each of our 169 TFs, we applied a linear regression with an interaction term to discover TF level—genotype interactions on gene expression for our filtered variants across 32,151 genes, selecting the top eQTL variant per gene for each analysis [31] (Fig 1E and 1F). Requiring that the TF-eQTL variant also had a significant eQTL signal in the respective tissue, we discovered 30 to 71,848 TF-eQTLs (eQTLs with TF interaction) per tissue at a 5% TF-level Benjamini-Hochberg (BH) FDR, with 276,394 relationships supported by at least one tissue (S5 Fig). These TF-eQTL pairs represent potential TF regulators of eQTL effects in those tissues.

We observed that five tissues (Whole Blood, Fibroblast, Colon, Stomach, and Testis) were outliers in the number of TF-eQTLs, which could not be explained by tissue sample size alone (S5 Fig). Analysis of *in silico* cell type estimates revealed that four of these tissues (Whole Blood, Fibroblast, Colon, and Stomach) had particularly high inter-individual variability in cell type composition (S6 Fig). Assuming that this high cell type composition variability was likely contributing to the large number of TF-eQTLs, we removed these tissues from our analysis so that our TF-eQTL results were not dominated by non-causal correlations of TFs with cell type composition. We also removed the Testis tissue due to its outlier status in previously reported gene expression and *trans*-regulation analyses [1,4], so that TF-eQTLs in this one tissue would not dominate the results.

Our final within-tissue dataset consisted of 40,065 TF-eQTL relationships supported by at least one tissue, of which 4,318 were supported by multiple tissues (Figs 2A and S7). Some TFs with many interacting eQTLs in a tissue made clear biological sense. For instance, PPARG, a key regulator of adipogenesis [32], was the TF with the most interactions in adipose tissue, and the TF with the most interactions in the nucleus accumbens was CEBPA, which is involved in neuronal functions [33] (Fig 2A). We see that 111/120 tissue pairs were enriched for one another's TF-eQTLs (OR = 12.1 to 1251, Fisher's exact test, all $p < 10^{-4}$) (Fig 2C). All those tissue pairs with an observed overlap depletion included a brain tissue and/or the lymphoblastoid cell line and did not have any overlapping TF-eQTL interactions (gray squares), likely due to

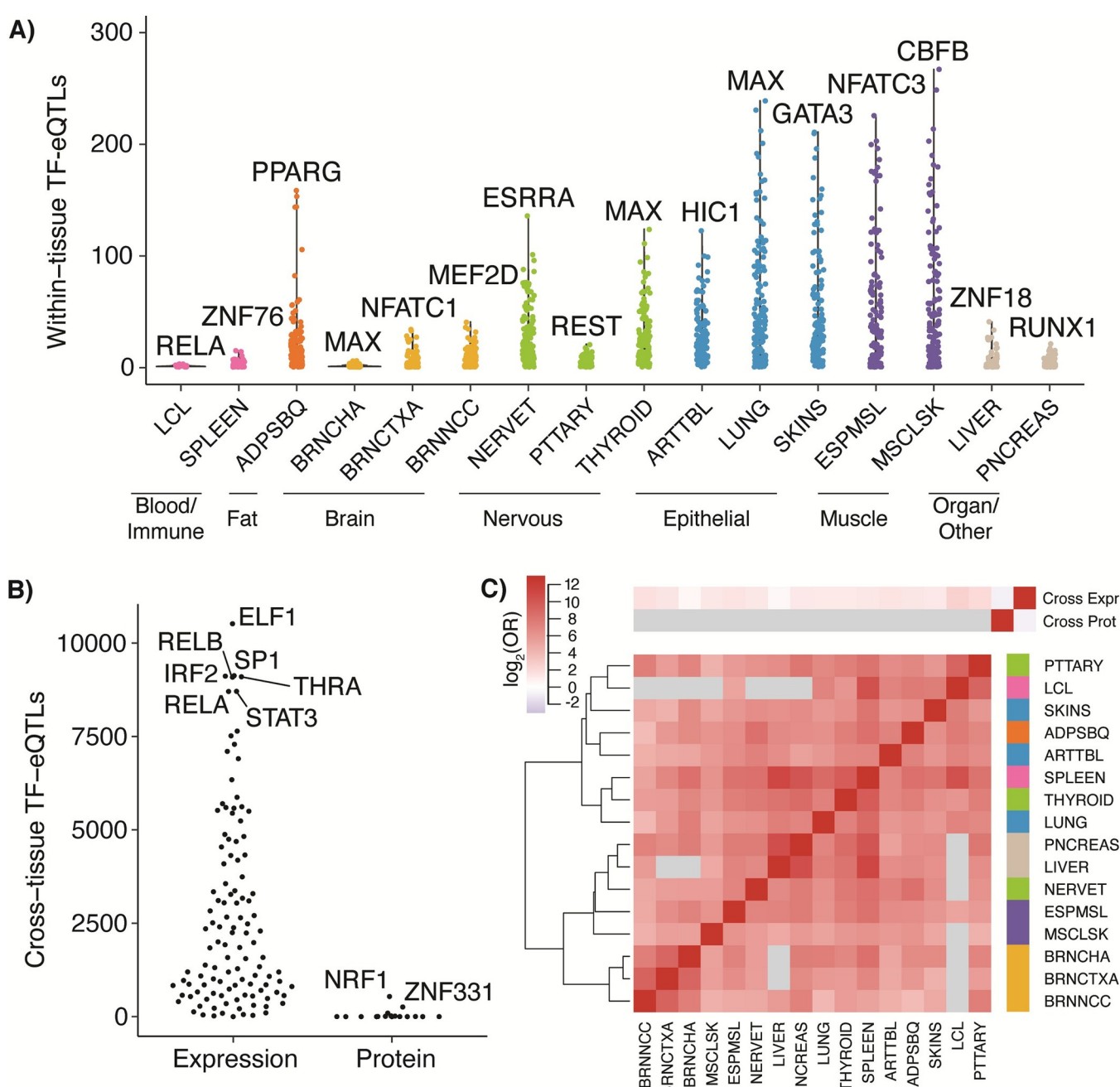

**Fig 2. Discovered TF-eQTL interactions. A)** Number of within-tissue TF-eQTL interactions at 5% FDR is plotted per TF for each tissue analyzed. The TF with the most interactions per tissue is highlighted. **B)** Number of discovered cross-tissue TF-eQTL interactions per TF for expression-based interactions and protein-based interactions (at 5% FDR). TFs with the most correlations per analysis are highlighted. **C)** Sharing of TF-eQTL interactions between tissues and with cross-tissue datasets. Red indicates positive enrichment and blue, negative enrichment. Grey squares indicate no shared TF-eQTL gene pairs between the two datasets.

the small sample sizes and these tissue types being highly distinct from others (all ORs = 0, Fisher's exact test ps = 1) [5]. In general, within-tissue TF-eQTL relationships follow a similar clustering pattern to tissue gene expression (Figs 2A and S4). These results highlight unique and shared potential TF regulators of eQTL effects within different tissue contexts.

## Correlation of eQTL effect sizes and TF levels across tissues

To obtain further insights into TFs driving eQTL effect size variation between tissues, we next investigated how TF levels across the 49 GTEx tissues correlated with eQTL effect sizes. We calculated log2 allelic fold change effect sizes (aFCs) in every GTEx tissue for each filtered variant-gene pair; by ignoring the original eQTL significance cutoffs and calculating aFCs in all tissues, we captured tissues lacking eQTL effects and avoided power differences in eQTL detection caused by varying tissue sample sizes. We correlated aFCs for each eQTL with expression levels for each of 169 TFs (Fig 1D) and selected the top eQTL variant per gene for each TF. We found 420,248 TF-eQTL correlations at a 5% TF-level BH FDR (Fig 2B). These TF-eQTL pairs represent potential TF regulators of eQTL effects that may explain the variability of these eQTLs across tissues. Many of the TFs with the most correlations in the cross-tissue analysis were involved in immune (ELF1, IRF2, RELB, STAT3, RELA) or hormone response (THRA) [34–38] (Fig 2B). Though we discovered many more potential TF-eQTL relationships across tissues than within tissues, the two sets of TF-eQTL interactions are enriched for one another (OR = 2.38, Fisher's exact test $p < 10^{-300}$), and cross-tissue correlations showed a positive direction of enrichment for all individual tissues (ORs >= 1.4) (Fig 2C).

Gene expression levels do not always directly correspond to protein levels [39,40], so we performed a similar correlation analysis using TF protein levels across tissues, as assessed by high-throughput mass spectrometry [41]. Protein quantification was available for 73/169 TFs in 20 or more tissues, with one to eleven samples per tissue (S1 Table). We discovered 965 TF protein-eQTL correlations across 943 genes at a 5% TF-level BH FDR (Fig 2B). These protein-based TF-eQTL correlations were depleted for expression-based cross-tissue TF-eQTLs (OR = 0.58; Fisher's exact test $p = 3.3x10^{-5}$) and had no overlap with any within-tissue TF-eQTLs (Figs 2C and S8). As discussed in Jiang et al., gene and protein levels may differ due to biological phenomena of RNA dynamics and translational regulation as well as technical variation in mass spectrometry technology that plagues especially lowly expressed proteins [41]. Given that TF protein levels are lower than other genes (Wilcoxon rank sum test $p < 10^{-300}$) (S9 Fig) and the number of assayed tissues and samples is small, these protein measurements may be less suitable measurements of TF levels for the purposes of this analysis.

## Annotation and TF-binding of TF-eQTL interactions

Next, we set out to evaluate our discovered sets of putative TF regulators of eQTLs, based on orthogonal data of functional annotations and TF binding. We examined four TF-eQTL datasets: cross-tissue expression-based, cross-tissue protein-based, within-tissue expression-based, and at least two lines of expression-based evidence (at least two tissues, or cross-tissue + at least one tissue). First, we examined genomic annotations of the top TF-eQTL variant for each gene and found that all three expression-based datasets were enriched to overlap promoters, 5' UTR, and 3' UTRs compared to all tested eQTL variants (S10 Fig). This is consistent with overall eQTL enrichments [1,4], suggesting that TF-eQTL variants are further enriched for true causal regulatory variants.

We tested whether our four sets of putative TF-eQTL interactions overlapped TF binding sites (TFBS) based on two datasets: ENCODE TF ChIP-seq peaks and HOCOMOCO predicted TF binding motifs. The top TF-eQTL variants showed enrichment for TF ChIP-seq overlap in all expression-based datasets and mixed results on TF motif matching enrichment (S11 Fig). Given the complicated structure of our data, with multiple variants tested per gene and LD between variants (S2 and S12 Figs), we set up a more sophisticated test of TFBS overlap enrichment of TF-eQTL interactions to account for this unusual data structure. We compared the observed overlap per gene to a null expectation and calculated significance using a

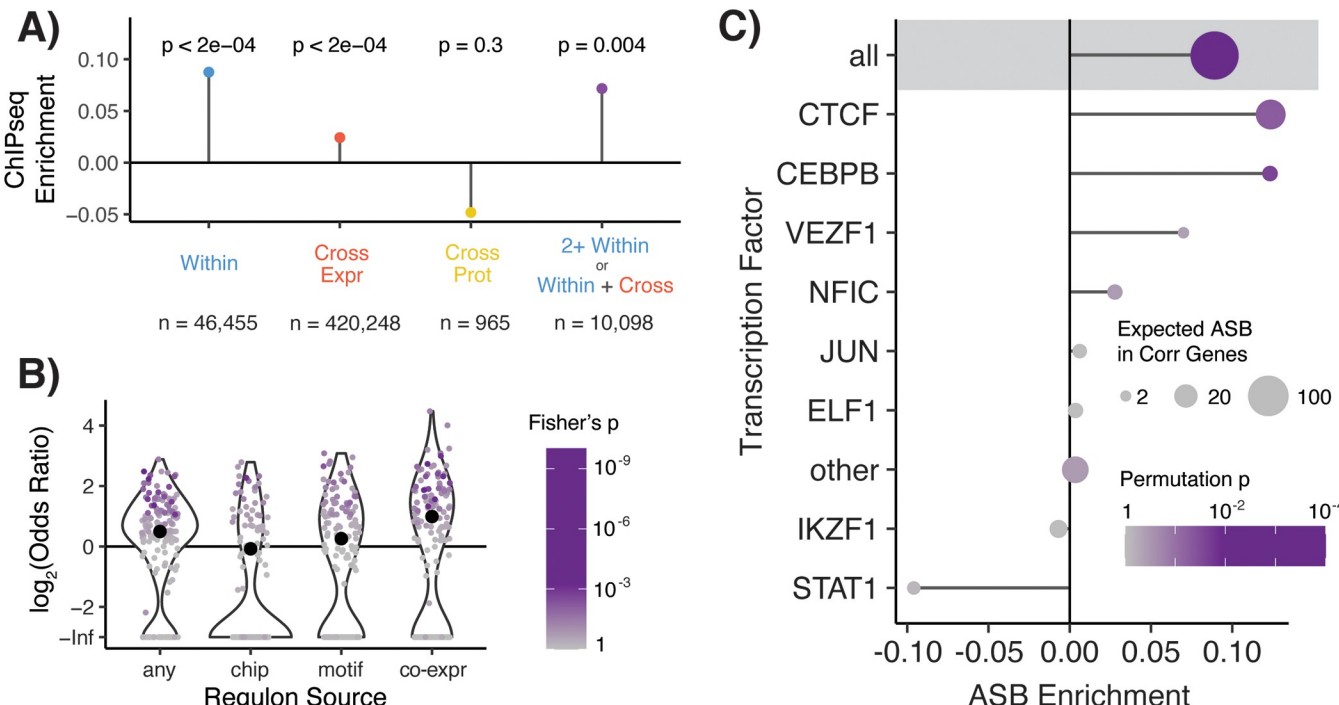

**Fig 3. TF binding of TF-eQTL interactions. A)** Overlap enrichment of TFBS, based on TF ChIP-seq peaks, of TF-eQTL interactions by dataset. Permutation-based p-values are plotted above each measurement. Datasets include within-tissue (blue) interactions, cross-tissue expression-based (red), cross-tissue protein-based (yellow), and TF-eQTL interactions with at least two lines of evidence from cross-tissue expression-based and within-tissue interactions (purple). **B)** The enrichment of target genes with two lines of evidence for TF-eQTL interactions falling into that TF's regulon. Large black dots depict overall enrichment across TFs. **C)** Enrichment for allele-specific TF binding (ASB) for TF-eQTL interactions with two lines of evidence. Shaded area contains statistics for unmatched TF ASB analysis. Below that, statistics for matched TF ASB analysis is shown, with TFs with more than one expected ASB event plotted individually, and all other TFs combined (other).

permutation scheme of TFBS overlap annotations (see *Methods*). Our enrichment statistic can be interpreted as the average number of extra variants with overlap per TF-eQTL gene. Both cross-tissue and within-tissue expression-based datasets were significantly enriched for ChIP-seq overlap (cross-tissue enrichment statistic = 0.024, $p < 2x10^{-4}$; within-tissue enrichment statistic = 0.088, $p = 2x10^{-4}$), and cross-tissue expression-based TF-eQTLs showed a small trend of motif matching enrichment (enrichment statistic = 0.002, $p = 0.06$) (Figs 3A and S13). TF-eQTLs with at least two lines of expression-based evidence were significantly enriched for ChIP-seq overlap (enrichment statistic = 0.072, $p = 4x10^{-3}$) (Figs 3A and S13). The cross-tissue protein-based TF-eQTL interactions had no significant enrichment for ChIP-seq overlap or motif matching (Figs 3A and S13); thus, taken together with their low concordance with expression-based interactions (Fig 2C), we decided not to pursue these interactions any farther.

Though our discovered expression-based TF-eQTL relationships were generated using only genetic and gene expression data, those eQTLs were more likely to overlap a TFBS of their interacting TF than expected (Fig 3A). We included all 10,098 TF-eQTL interactions with at least two lines of expression-based evidence to represent a high-confidence set of putative TF regulators of genetic variant effects. These TF-eQTL genes were also enriched to fall into the regulon of the interacting TF (any regulon set OR = 1.41, Fisher's exact test $p = 3x10^{-17}$) [42], with the strongest enrichment seen for regulons defined by co-expression analysis (OR = 1.99, Fisher's exact test $p = 8x10^{-30}$) (Fig 3B). These 10,098 dual-evidence (DE) TF-eQTL

interactions, observed across 154 TFs and 2,136 genes, represent potential TF regulators of genetic variant effects (S2 Table) that we then analyzed further.

## Allele-specific TF binding of dual-evidence TF-eQTL interactions

We next examined TF ChIP-seq allele-specific binding data to determine if our dual-evidence (DE) TF regulators of genetic variant effects manifested altered TF binding *in vivo*. To accomplish this, we used the ADASTRA dataset, which contains allele-specific TF binding (ASB) results from over seven thousand TF ChIP-seq experiments, normalized for cell-type-specific background allelic dosage [43]. Like our TFBS overlap enrichment analysis, we compared the observed allele-specific TF binding of DE TF-eQTL interactions to a null expectation, followed by permutation of ASB annotations to estimate the enrichment significance.

We observed that DE TF-eQTL variants were significantly more likely to have ASB in general, with any TF (enrichment statistic = 0.089, p = $4\times10^{-4}$) (Fig 3C). Testing for the enrichment of ASB for the matching TF-eQTL TF was limited by the sparsity of the ASB data: only 16 out of 90 analyzed TFs were expected to have more than one interacting TF-eQTL with an ASB event (S14 Fig). However, we were able to observe modest but significant enrichment for ASB of the implicated TF when combined across TFs (enrichment statistic = 0.016, p = $7\times10^{-3}$). These results demonstrate that our DE TF-eQTL interactions are enriched for variants that alter TF binding, including for the specific TF of the TF-eQTL.

## IRF1 knockdown validates IRF1-eQTL interactions

Our dual-evidence TF-eQTLs included 87 eQTL effects putatively regulated by IRF1, which we assessed further with an IRF1 knockdown experiment. We used data from a CRISPR-interference-mediated knockdown of IRF1 in HEK293-TLR4 cells [44] and measured genes' allele-specific expression (ASE) at knocked-down and control IRF1 levels (Fig 4A and 4B). A change in ASE between IRF1 conditions would suggest that IRF1 is regulating the effect of the heterozygous eQTL on gene expression.

We compared allele-specific gene expression in IRF1-knockdown and control cells, combining reads across all samples per condition to increase our power to discover differences in allelic expression. After filtering for sufficient coverage of a heterozygous coding SNP in the HEK293T samples (>60 reads, >5% REF reads, >5% ALT reads, and <5% non-REF/ALT reads), we were left with 1,221 genes for which we performed Fisher's exact test for imbalanced allelic expression across conditions. A low Fisher's test p-value indicates that the two alleles are expressed at different ratios in the knockdown and control conditions, suggesting that IRF1 controls the expression of the gene in an allele-specific manner in this cell line.

We discovered 87 nominally significant genes with differing ASE between IRF1 conditions (Fisher's exact test p < 0.05). Of the eleven dual evidence IRF1-eQTL genes with measurable ASE, eight were heterozygous for an implicated top IRF1-eQTL variant, thus we could expect them to show differing ASE between IRF1 conditions. Indeed, *ERI1*, *MYOM2*, and lncRNA *RP5-1092A3.4* showed nominally significant conditional ASE (Figs 4E, S15, and S16). Furthermore, all eleven DE IRF1-eQTL genes with a heterozygous IRF1-eQTL variant in HEK293T had a coding variant with a p value in the lower quartile of tested genes, and none of the three DE IRF1-eQTL genes without a heterozygous variant showed significant conditional ASE (S15 Fig). Examining ASE in this IRF1 knockdown experiment validated 3/8 of our testable IRF1-eQTL interactions and demonstrates the high promise of this method to generate useful TF regulation information that can be applied to understand allele-specific regulation in new contexts.

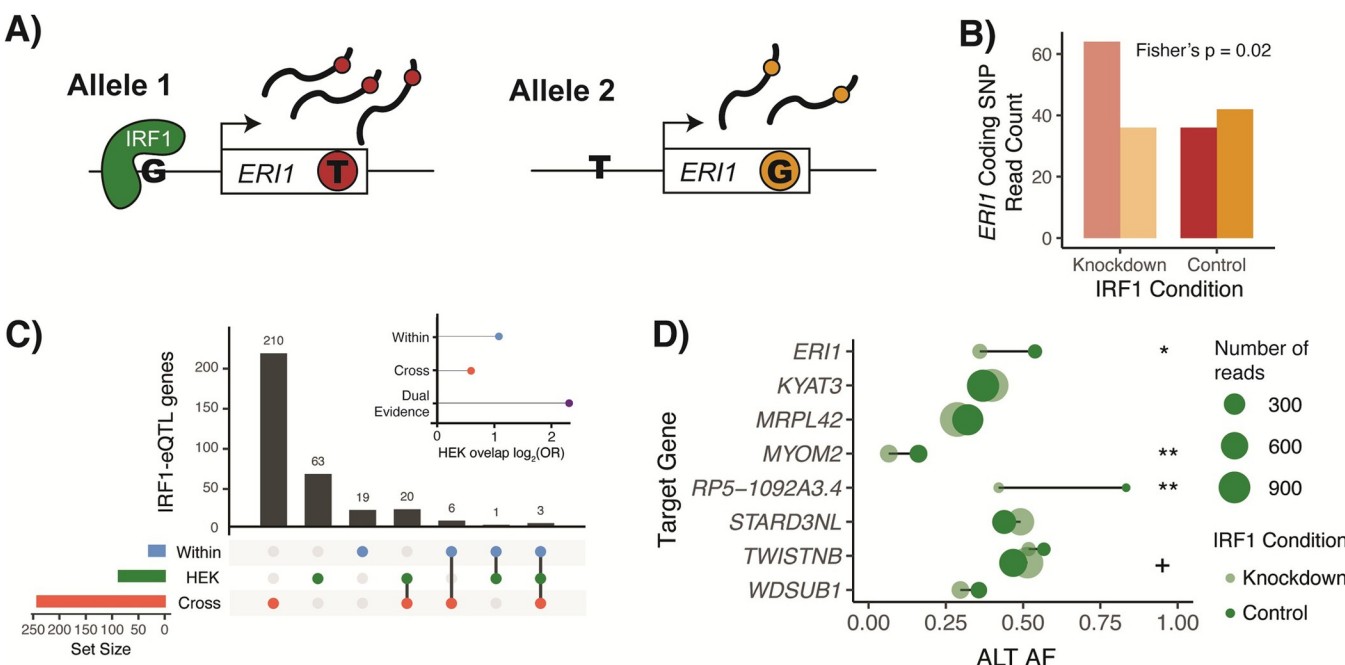

**Fig 4. IRF1-eQTL interactions in HEK293-TLR4 IRF1 knockdown. A)** Depiction of allele-specific expression, with IRF1 preferentially binding to the G-allele in the regulatory region of the ERI1 target gene. This leads to higher expression of allele 1, which we can measure based on the presence of a heterozygous coding SNP in the ERI1 transcript. Reads from allele 1 will have a T genotype (red) at the coding SNP and reads from allele 2 will have a G (orange). **B)** Read counts for ERI1 coding SNP alleles in both knockdown and control conditions. In this example, it appears that we observe allelic effects at lower (knockdown) IRF1 levels, while higher (control) levels of IRF1 may saturate binding to both alleles. Conditions are compared using Fisher's exact test of allelic counts. **C)** Sharing of IRF1-interacting eQTL genes in within-tissue (blue), cross-tissue expression-based (red), and HEK293T IRF1 knockdown (green) datasets. Only genes with an adequately expressed heterozygous coding SNP in HEK293T samples are included. Inset shows enrichment for overlap between HEK293T IRF1-eQTL genes and listed datasets. **D)** HEK293T coding SNP alternative allele frequency in dual-evidence IRF1-eQTL genes that were heterozygous for a top TF-eQTL variant and had adequate coverage of a heterozygous coding SNP. + indicates a Fisher's p value $< 0.1$, * $< 0.05$, ** $< 0.01$ of allelic counts vs. condition.

## TF regulation of gene-by-environment effects and genetic effects on phenotype

We hypothesized that our TF-eQTLs could shed light on mechanisms of gene-by-environment (GxE) interactions that represent environmental conditions that affect genetic control of a phenotype, and altered TF level could be the mechanism by which the environmental condition regulates the genetic effects. In a recent large-scale study, Findley et al. tested the effects of 14 environmental treatments on allele-specific gene expression in three cell lines, discovering 979 genes with GxE effects, 850 of which were also found to have a GxE interaction by a previous study [12]. We overlapped our DE TF-eQTL genes with these 850 replicated GxE interacting genes and found 114 overlaps (OR = 2.43, Fisher's exact test p = $9\times10^{-15}$) (Fig 5A), which offer potential direct mechanistic interpretations of the environmental effects on genetic control of gene expression (S3 Table). For instance, we found multiple GxE interactions for dexamethasone treatment that overlapped DE TF-eQTL genes for RELA and RELB [45]. Dexamethasone is a known inhibitor of NF-κB signaling (of which RELA and RELB are members), suggesting that they could be the mechanism by which dexamethasone regulates genetic effects at these loci. Thus, combining TF-eQTL mechanisms with GxE interactions has the potential to elucidate direct mechanisms of environmental effects.

We next assessed if we could use our DE TF-eQTLs to discover TF regulators of GWAS loci. Colocalization methods can combine statistical signals from eQTLs and GWAS loci to

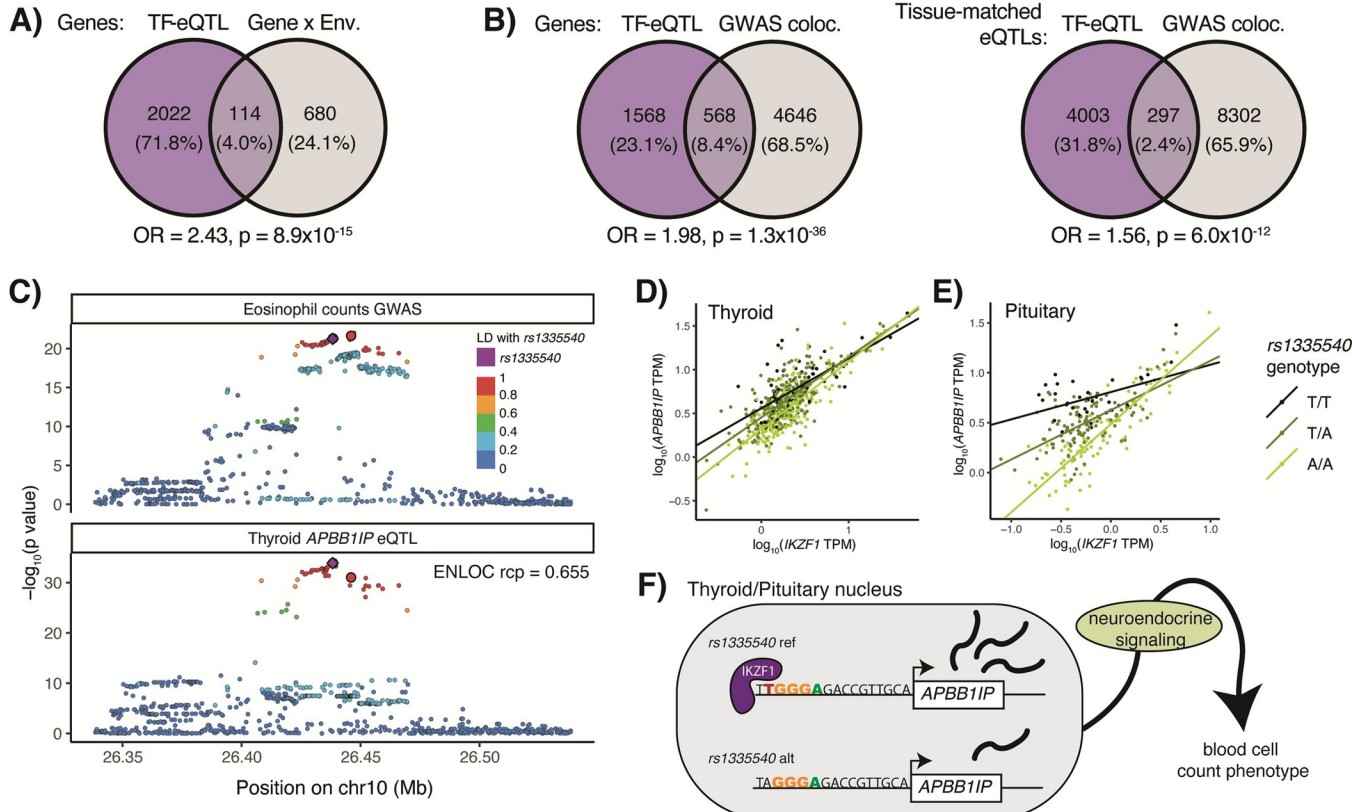

**Fig 5. TF-eQTL implications for gene-by-environment and GWAS effects. A)** Overlap of TF-eQTL genes with GxE genes from Findley et al. 2021. **B)** Overlap of TF-eQTL genes with GWAS colocalizing eQTL genes from GTEx. The first diagram shows overlap for a gene with a TF-eQTL in any tissue and colocalizing eQTL in any tissue. The second shows overlap of tissue eQTLs with TF-eQTL and/or colocalizing GWAS locus in the given tissue. **C)** Representative eQTL and GWAS p-values are plotted for variants in the region of an APBB1IP eQTL and blood trait GWAS locus. Lead variants from IKZF1-eQTL interactions in thyroid, pituitary, and tibial artery are larger and outlined in black. (The lead variant from pituitary/artery cannot be seen as it falls behind rs1335540.) **D) & E)** Individual samples in thyroid and pituitary tissues are plotted by IKZF1 and APBB1IP expression, and linear regression lines are plotted by genotype. The difference in APBB1IP expression between the genotypes gets smaller as IKZF1 expression increases across the samples. **F)** Schematic of IKZF1 regulation of APBB1IP and blood cell counts. An IKZF1 binding site predicted by the HOCOMOCO IKZF1 motif lies nine bases upstream of APBB1IP's transcription start site, which is disrupted by the alternative allele of rs1335540. Under our neuroendocrine signaling hypothesis, APBB1IP expression in neuroendocrine tissues goes on to alter system-wide neuroendocrine signaling, which would cause changes in blood cell counts. As IKZF1 appears to regulate the APBB1IP eQTLs in these tissues, it would follow that IKZF1 TF therefore may regulate the effect of this locus on blood cell counts.

determine if the gene and phenotype regulation share a causal variant, implying that genetic regulation of the gene may be the causal mechanism of the genetic effect on phenotype. We obtained GWAS-eQTL colocalization data of GTEx eQTLs and 76 GWAS traits [46,47] and combined these with our DE TF-eQTLs. We saw that our DE TF-eQTL genes were more likely to colocalize with GWAS loci than all tested eQTL genes, with 27% of our DE TF-eQTL genes showing colocalization between a GWAS signal and an eQTL in any tissue and 6.9% showing colocalization between a GWAS signal and an eQTL in the same tissue where the TF-eQTL was discovered (OR = 1.98, 1.6, Fisher's exact test p = $1x10^{-36}$, $6x10^{-12}$, respectively) (Fig 5B). We found 1,212 colocalizations between a GWAS signal and an eQTL signal in a tissue with a DE TF-eQTL that had high LD between the lead colocalization and TF-eQTL variants ($r^2$ > 0.4), which represent potential TF regulators of genetic effects on phenotype (S3 Table).

One example of this relationship is an *APBB1IP* eQTL that interacts with transcription factor IKZF1. This eQTL is present in 31 GTEx tissues and colocalized with GWAS signals for four red and white blood cell traits (ENLOC regional conditional probability > 0.5),

suggesting that genetic control of these traits could be mediated by *APBB1IP* expression [46–48] (Figs 5C and S17 and S4 Table). We observed three tissues (pituitary gland, thyroid, and tibial artery) with TF-eQTL interactions for the *APBB1IP* gene with IKZF1 (Figs 5D, 5E, and S18). Supporting IKZF1 regulation of this eQTL, the top TF-eQTL and GWAS variants were highly linked ($r^2 > 0.85$) to *rs1335540*, a SNP found 15 bases upstream of an *APBB1IP* transcript start site that overlaps an IKZF1 ChIP-seq peak and matches a IKZF1 motif [49–51] (Figs 5F, S18, and S19). *APBB1IP* eQTLs in all three tissues with an IKZF1-eQTL interaction showed colocalization with blood cell traits. *APBB1IP* mediates blood cell adhesion and immune response [52,53]. It is also involved in integrin-mediated changes in the actin cytoskeleton of mammalian cells [54,55] and its orthologue MIG-10 has been shown to regulate axon outgrowth in *C. elegans* neurons [56]. IKZF1 is a chromatin-remodeling TF involved in lymphocyte development as well as the neuroendocrine system [57,58]. These findings offer two explanations for the genetic control of blood cell traits by *APBB1IP* expression: 1) via altered gene expression in the blood cells themselves, or 2) via neuroendocrine control of blood cell counts originating with altered gene expression in neurons. Offering further support to the neuroendocrine hypothesis, thyroid dysfunction has been shown to alter red and white blood cell counts [59,60], and the IKZF1-eQTL interactions were observed in neuroendocrine tissues. Regardless, given the shared genetic signal in multiple tissues, we can hypothesize that IKZF1 regulates both *APBB1IP* expression and the implicated blood traits, suggesting a TF regulator of a complex trait's genetic association.

## Discussion

In this paper, we used the natural variation of TFs across tissues and individuals to discover 10,098 TF-eQTL interactions across 2,136 genes, which represent putative TF-based mechanisms of genetic effects on gene expression. These TF-eQTLs were supported by at least two lines of evidence, including cross-tissue and/or within-tissue variation. They were enriched to overlap ChIP-seq peaks and fall into the regulon of the implicated TF, corroborating with orthogonal evidence that these eQTLs are regulated by the implicated TFs. Furthermore, analysis of an IRF1-knockdown experiment validated three out of eight testable IRF1-eQTLs. We see that TF-eQTL genes are more likely to colocalize with GWAS loci and overlap genes with gene-by-environment effects, and our example of IKZF1 regulation of an *APBB1IP* eQTL that colocalizes with GWAS signals for blood cell traits illustrates how our TF model can be used to discover likely TF regulators of GWAS effects.

Given the high number of possible causal genetic variants and putative regulatory mechanisms based on statistical fine-mapping and functional annotation overlap, it is clear that additional methods are needed to pinpoint causal variants and mechanisms of quantitative trait loci. Our method offers a new approach to discovering TF regulation of a genetic variant's effects, which can help us determine the eQTL's potential mechanism of action and explain its context variability. One major advantage to our method is its accessibility. While functional annotations were used to choose variants to test and to validate our results, the main discoveries of the model were powered by sample genotypes and gene expression levels–the same data available in most eQTL analyses. We leverage variation cross-tissues and within-tissues, which both have value for discovering TF regulators of eQTL effects, but especially the within-tissue TF interaction analysis is applicable to any eQTL data set even when a large number of different conditions may not be available.

Understanding TF regulation of an eQTL effect can allow us to focus functional fine-mapping efforts only on the implicated TF, hopefully narrowing the focus to one or a handful of

variants that disrupt binding sites predicted by that TF's motif or show allele-specific binding in its ChIP-seq data. Unlocking these mechanisms allows us to eventually improve our understanding of the regulatory code of the genome and how human genetic variation perturbs that system. One clear application of our approach is for discovering and interpreting gene-by-environment (GxE) effects on gene expression and phenotype. While GxE interactions on human phenotype have been difficult to assess, GxE interactions in relation to gene expression have been studied under various contexts [8,12–15,61–63]. Overlapping these effects with TF-eQTLs as in our analysis, or even performing TF-eQTL analysis in the environmental exposure datasets themselves, provides mechanistic hypotheses of how environmental effects impact genetic control of gene expression and phenotype.

We were surprised by the lack of validation of TF-eQTLs discovered with cross-tissue protein levels, since protein measurements should reflect TF activity levels more accurately than expression measurements. However, the protein data had less power, from a smaller number of individuals and tissues than the expression data, and mass spectrometry may have more technical noise than expression quantifications from RNA-seq. Another promising option for TF measurement in the model is TF activity as predicted by target gene expression [64], which should account for translation rates, post-translational modifications, and subcellular localization effects on TF activity that expression measurements cannot capture. Initial analyses with this datatype did not yield strong results, but as activity estimates improve, the option should be revisited.

Though we saw enrichment for TF ChIP-seq peaks and allele-specific binding, our TF binding enrichments were quite modest. For instance, the TF ChIP-seq overlap enrichment statistic of 0.07 for dual-evidence TF-eQTLs means that we observed 0.07 more variants with ChIP-seq overlap per TF-eQTL gene than expected–or one additional overlap per twenty genes. Part of this may arise from the lack of ChIP-seq data from relevant tissue and cell type contexts that match the GTEx eQTL data. Nonetheless, it is likely that our dual-evidence TF-eQTLs likely contain false positives. One of the factors that may contribute to this is the correlated expression between TFs, which is difficult to fully account for. Another important factor is TF-eQTL correlations that may be caused by cell type composition [6], such that an eQTL only found in a given cell type might be correlated with TFs that are highly expressed in that cell type even when the TF does not specifically regulate the eQTL. While some of our discovered TF-eQTLs may be false positives due to cell type variability, the ChIP-seq enrichments and IRF1 validation indicate that the applied filters successfully remove many of the major cell type composition effects. Altogether, we consider our 10,098 TF-eQTLs to represent regulatory variants with an indication of being regulated by the implicated TFs, but full validation will require additional work.

In summary, in addition to this catalog of potential TF regulators of eQTLs, we hope that our methods of comparing TF level with genetic variant effect can be applied in additional eQTL datasets, as well as for splicing QTLs and other molecular phenotypes. Our approach has the potential to implicate mechanisms for eQTL effects that vary across contexts without requiring additional datatypes or experiments, though its integration with other lines of evidence can further strengthen the insights, as shown in this study. Additionally, our method can improve functional fine-mapping efforts by highlighting TFs that may be regulating a locus, which can be further investigated with functional genomic data for that TF such as motif prediction or allele-specific binding data. We believe this TF-based framework of genetic variant effect variability can advance our understanding of QTL and GWAS mechanisms and their context variability, with great promise for understanding environmental interactions that impact genetic disease risk.

## Methods

### GTEx data

For the bulk of our analysis, we used the GTEx v8 dataset, including whole genome sequencing for 838 individuals and mRNA sequencing from 15,201 samples across 49 tissues (Tables 2 and S1). RNA-seq data were aligned using STAR v2.5.3a, and gene counts were based on GEN-CODE Release 26 and analyzed using RNA-SeQC [5]. *cis*-eQTL calculations in each tissue and Caviar fine-mapping 95% confidence sets for those eQTLs were also previously generated (Table 2) [5]. High-throughput mass spectrometry protein measurements were separately available for 201 GTEx samples across 32 tissues [41] (Tables 2 and S1). GTEx tissues were categorized into Blood/Immune, Adipose, Brain, Nervous System (non-brain), Epithelial, Muscle, or Organ/Other via a cursory literature search on biological composition and function.

### Filtering variants

We limit our analysis to variants where we have prior evidence to suggest that this could be a variant affecting gene expression that is regulated by a TF. We filtered for variants that matched four criteria: 1) $>= 5\%$ minor allele frequency in GTEx v8 samples; 2) present in a Caviar fine-mapped 95% credible set for an eQTL in any GTEx tissue; 3) overlap an ENCODE TF ChIP-seq peak for at least one of 169 TFs; 4) match a HOCOMOCO consensus sequence motif for at least one of 169 TFs. We used ENCODE narrowPeak regions in all available experiments that passed filtering criteria (as of January 2020) and HOCOMOCO v11 IUPAC consensus motifs (Table 2). For the ENCODE TF ChIP-seq overlap, we used ChIP-seq optimal irreproducible discovery rate (IDR) threshold peak files for experiments with a biological replicate, no red or orange audit categories, and no experimental conditions. We used a union of regions if multiple IDR files were available per TF. For HOCOMOCO TF motif matching, we converted the IUPAC consensus sequence motif to a regular-expression string for both the forward and reverse-compliment motif, trimming any less confident bases (degenerate to three or four nucleotides) from the ends of the sequence. We extracted the genomic sequence surrounding each variant (motif length minus one on either side of the variant) using samtools, and we used grep to check if the forward or reverse-compliment motif was present in the reference and/or the alternative alleles. The above filtering left us with 473,057 variants. Using the Caviar fine-mapping data, we associated each filtered variant with one or more eGenes, which resulted in 1,032,124 eVariant-eGene pairs across 32,151 genes. Unless otherwise stated, all enrichment testing was performed with these filtered variants and genes as the background set.

**Table 2. Data sources.**

| Data Type | Publication DOI / Citation | Website |
|---|---|---|
| GTEx v8 genetic, gene expression, eQTL, and fine-mapping data | 10.1126/science.aaz1776 / [5] | https://gtexportal.org/home/ |
| GTEx protein data | 10.1016/j.cell.2020.08.036 / [41] | |
| GTEx GWAS colocalization | 10.1186/s13059-020-02252-4 / [47] | |
| ENCODE TF ChIPseq peaks | Multiple experiments | https://www.encodeproject.org/ search/?type=Experiment |
| HOCOMOCO TF motifs | 10.1093/nar/gkx1106 / [50] | https://hocomoco11.autosome.ru/ |
| TF regulon sets | 10.1101/gr.240663.118 / [42] | |
| ADASTRA allele-specific binding data | 10.1038/s41467-021-23007-0 / [22] | https://adastra.autosome.ru/susan |
| HEK293-TLR4 IRF1 knockdown experiment | 10.1101/2020.02.21.959734 / [44] | |
| HEK293-TLR4 genome sequence | 10.1038/ncomms5767 / [65] | http://hek293genome.org/v2 |
| Gene-by-environment interactions | 10.7554/eLife.67077 / [12] | |

## Within-tissue interactions

For our within-tissue TF-eQTL interaction discovery, we selected twenty tissues that best represented all 49 GTEx eQTL tissues based on gene expression clustering. We clustered tissues based on median TPM across all genes using Euclidean distances and Ward.D clustering, cut the resulting tree to generate twenty clusters, and selected the tissue with the largest sample size from each cluster. If a tissue was removed for cell type composition variability (below), the next largest tissue was selected from the cluster, if one was available.

For each selected tissue, we applied an eQTL interaction model to discover TF-eQTL interactions on gene expression. We ran tensorQTL software per TF and per tissue for 32,151 genes and all variants within a 10 mega-base window of the transcription start site, inputting individuals' genotypes, normalized eGene expression, and normalized TF expression for each eQTL-TF pair, as well as genotype principal components and tissue covariates described in The GTEx Consortium, 2020 [5]. We then selected the filtered variant sets for each gene and calculated a corrected p-value using the effective number of independent variants tested per gene. We defined the effective number of tests per gene as the number of eigenvectors needed to capture 95% of the variance in the GTEx genotype matrix of all tested variants, using the Gao method in the poolr package [66]. We then applied a Benjamini-Hochberg (BH) correction to the meff-corrected p-values across each tissue and TF. For all TF-eQTL interactions with BH FDR $< = 5\%$, we selected those where the top TF-eQTL variant had a significant eQTL signal in the respective tissue and where the gene was not the implicated TF.

We removed four tissues with high cell type composition variability so that our results were not dominated by non-causal TF-eQTL relationships due to cell type composition (Whole Blood, Fibroblast, Colon, Stomach), and we removed one tissue due to its high number of results and unique gene expression patterns (Testis). Cell type composition was estimated in Kim-Hellmuth et al., 2020 [6]: briefly, XCell was used to calculate enrichment of cell-type-specific gene expression signals in GTEx samples [67]. Since these estimates were not all experimentally validated, we ignored cell type estimates with high variability across tissues (aDC, iDC) (S6 Fig). Four cell type estimates had high variability in a GTEx tissue (variance $> 0.04$; Th2 cells in fibroblasts, epithelial cells in the colon, epithelial cells in the stomach, and basophils in blood) and those tissues were removed from the analysis to avoid strong cell type interaction signals in our results. Stomach clustered with other tissues (S4 Fig), so we added the next largest tissue in that group, Pancreas, to our within-tissue TF-eQTL analysis.

## Cross-tissue correlations

We correlated eQTL effect sizes and TF expression levels across up to 49 GTEx tissues. We used the aFC software package to calculate eQTL effect sizes based on log2 allelic fold change (aFC) for all 1,032,124 filtered eVariant-eGene pairs in each tissue [68], using genotype principal components and tissue covariates described in The GTEx Consortium, 2020 [5]. We determined the median TF level per tissue based on transcripts per million (TPM). Then we performed a cross-tissue Spearman correlation of eQTL aFC and TF median TPM for each eQTL-TF pair in all tissues with median eGene expression greater than 0 TPM, i.e., in all tissues where the eQTL target gene was sufficiently expressed. We tested 1–249 eVariants per eGene, and we selected the top variant per gene and calculated a corrected p-value using the effective number of independent variants tested per gene. We defined the effective number of tests per gene as the number of eigenvectors needed to capture 95% of the variance in the GTEx genotype matrix of all tested variants, using the Gao method in the poolr package [66]. We then performed a Benjamini-Hochberg TF-level correction of the meff-corrected p-values

across the top variants of each gene, and we selected variants with up to a 5% false discovery rate (FDR).

For our protein-based analysis, we used a similar approach, substituting TF protein levels for TF expression levels. We examined median protein levels in 32 GTEx tissues using normalized high-throughput mass spectrometry data [41]. We filtered for TFs with at least 20 unique protein values across tissues, then performed a cross-tissue Spearman correlation of eQTL aFC and TF median protein level. We tested our 1,032,124 filtered eVariant-eGene pairs and 72 TFs using the same p-value calculation procedure described above, then selected variants with up to a 5% FDR.

## Dataset comparison

We tested for TF-eQTL sharing across multiple datasets: cross-tissue expression-based, cross-tissue protein-based, and each within-tissue expression-based dataset. We performed Fisher's tests based on every TF-eGene pair's presence in the significant interactions from each dataset. For comparisons with cross-tissue protein-based data, we only used TF-eGene pairs for 72 TFs tested in the protein data.

## TF binding overlap enrichment

We tested whether our predicted TF-eQTL interactions overlap TF binding sites (TFBS) based on two orthogonal datasets: ENCODE TF ChIP-seq peaks and HOCOMOCO predicted TF binding motifs (Table 2). Given the complicated structure of our data, with multiple variants tested per gene and LD between variants, we used an expectation/observation model to test TFBS overlap enrichment of TF-eQTL interactions.

For each TF ($f$), we calculated the number of expected overlaps per gene ($g$) based on the number of variants tested ($v$) and the probability that any variant overlapped that annotation ($p$), and compared that to the observed number variants that overlap the annotation ($o$):

$$S_{f,g} = obs_{f,g} - exp_{f,g} = o_{f,g} - v_g * p_f \qquad \text{(Eq 1)}$$

We then averaged the per gene statistics across all genes with a significant TF-eQTL interaction ($G_f$):

$$S_f = \frac{\sum_{g \in G_f} S_{f,g}}{|G_f|} \qquad \text{(Eq 2)}$$

And we averaged across all 169 TFs to get our final enrichment statistic:

$$S_{overlap} = \frac{\sum_{f=1}^{169} S_f * |G_f|}{\sum_{f=1}^{169} |G_f|} \qquad \text{(Eq 3)}$$

The resulting statistic can be interpreted as the average extra number of overlaps per gene. For instance, an overlap enrichment statistic of 0.01 would mean that we observed 0.01 more variants with overlap per TF-eQTL gene than expected–or one additional overlap per 100 genes.

Permutations were carried out by shuffling overlap annotations across all tested variants and recalculating the overlap statistic $10^4$ times. Permutation p-values were calculated by counting the number of times the permuted TF statistic is larger or smaller than the observed statistic, adding one, dividing by the number of permutations, and multiplying by two for a two-sided test.

## Regulon enrichment

We examined whether our DE TF-eQTL genes were enriched to fall in regulons (target genes) of the implicated TF. We used four types of regulons defined by Garcia-Alonso et al., 2019 [42]. Curated meant that the authors examined published literature to find TF target genes; ChIPseq regulons were defined by TF ChIPseq peaks in the promoter of the gene; motif regulons were determined by TF motif matching in gene promoters; and co-expression regulons were determined by examining co-expression of transcription factors and target genes in GTEx tissues. The "any" regulon set was composed of a union of all other regulon types. We used Fisher's exact test to examine the overlap of DE TF-eQTLs and regulon sets for each TF and combined the results for each regulon set type.

## Allele-specific TF binding validation

We examined TF allele-specific binding (ASB) data to determine if our high-confidence set of potential TF regulators led to altered TF binding *in vivo*. We based our analysis on the ADA-STRA dataset (Susan version), which contains a meta-analysis of allele-specific TF binding results from over 7,000 TF ChIP-seq experiments (Table 2) [22]. Similar to our TFBS overlap enrichment analysis, we used an expectation/observation model to test allele-specific binding of TF-eQTL interactions, then permuted allele-specific binding annotations to calculate the enrichment significance.

For the un-matched TF ASB overlap analysis, we calculated the number of expected variants with ASB per gene based on the number of tested variants that were assayed in the ASB dataset ($v$) and the probability that any variant had ASB for any TF ($p$). We then compared the expected to the observed number of variants with ASB ($o$) in each gene, and we averaged across all genes that had a significant TF-eQTL interaction for any TF ($G_{any}$):

$$S_{any} = \frac{\sum_{g \in G_{any}} S_{\cdot g}}{|G_{any}|} \tag{Eq 4}$$

For our matched TF analysis, we calculated the number of expected variants with ASB per gene based on the number of tested variants that were assayed in the ASB dataset ($v$) and the probability that any variant had ASB for the specified TF ($p$). We then compared the expected to the observed number of variants with ASB ($o$) using the equation for $S_g$ described previously (Eq 1), then averaged the per gene statistics across all correlated genes using the equation for $S_f$ (Eq 2).

The overall enrichment was calculated using $S_g$ and $S_f$, with genes with a significant TF-eQTL interaction per TF ($G_f$) and 124 total TFs, using the full equation:

$$S_{matched} = \frac{\sum_{f=1}^{124} \sum_{g \in G_f} o_{f,g} - v_g * p_f}{\sum_{f=1}^{124} |G_f|} \tag{Eq 5}$$

We then permuted ASB annotations across all tested variants and recalculated the ASB statistic $10^4$ times. We calculated permutation p-values using the same two-sided test procedure described in our TFBS overlap enrichment analysis.

## IRF1 knockdown analysis

Our high confidence set of potential TF regulators included 58 eQTL effects predicted to be regulated by IRF1. To test these, we used a CRISPR-i knockdown of IRF1 in TLR4-expressing HEK cells (HEK293T) and measured allele-specific expression (ASE) at varying IRF1 levels

(Table 2) [44]. If we observe that ASE changes with IRF1 levels, this would suggest that IRF1 is truly regulating the effect of the eQTL on gene expression. First, we filtered the aligned HEK293T RNAseq data for coding variants that had adequate coverage to call ASE: at least 60 reads across all conditions, at least 5% reference allele and 5% alternative allele, and less than 5% of other alleles. Then, we used Fisher's test to compare the allelic balance across all promoter knockdown samples and all control samples. As our test was likely underpowered, we looked at genes with a 0.05 nominal p-value cutoff. We then checked which IRF1-eQTL top variants were heterozygous in the HEK293T cell line using VCF files from Complete Genomics (Table 2) [65]. All seven testable IRF1-eQTL were heterozygous for a top IRF1-eQTL variant in the HEK293T cell line.

## Comparison with GxE genes

Gene-by-environment interaction analysis results were attained from S4 Table in Findley et al., 2021 (Table 2) [12]. We matched these results by ENSG number with our dual-evidence TF-eQTL genes and used Fisher's exact test to calculate overlap compared to all tested genes in our dataset (i.e., all genes from filtered eVariant-eGene pairs).

## GWAS colocalization

To discover TF regulators of GWAS loci, we examined colocalization of GTEx eQTLs and 76 GWAS traits. We obtained ENLOC colocalization results (regional conditional probability > 0.5) from the GTEx Consortium (Table 2) [46,47] and overlapped these eQTL genes with our high confidence TF-eQTL genes. To test if our TF-eQTLs were enriched to colocalize with GWAS signals, we looked at all significant eQTL genes in any tissue and performed a Fisher's exact test for whether or not the gene had an eQTL that colocalized with a GWAS phenotype and whether or not we found a TF-eQTL interaction for that gene. We also performed a tissue-specific analysis where we looked at all significant eQTL genes in the 16 tissues where we performed within-tissue TF-eQTL discovery, and we performed a Fisher's exact test for whether or not the tissue's eQTL signal colocalized with a GWAS phenotype and whether or not a high confidence TF-eQTL was found for that gene in the tissue. Our background gene sets for both enrichment analyses included only genes all from filtered eVariant-eGene pairs. Our list of 1,212 colocalizing GWAS-eQTLs with a TF-eQTL were based on this tissue-specific comparison and were additionally filtered such that the $r^2$ of the top TF-eQTL variant and the lead ENLOC colocalizing variant was great than 0.4.

## Supporting information

**S1 Fig. Possible models of TF mechanisms of eQTL effect variability.** (Top) An eQTL variant increases the affinity of the purple TF, resulting in observable eQTL effects at mid-range levels of purple TF. (Middle) An eQTL variant disrupts the affinity of the gray TF, while the purple TF binds to a different region of the *cis*-regulatory region which interacts additively with the region where the gray TF binds. The eQTL effect will be observable at low purple TF levels, but may become overpowered and unobservable if purple TF levels increase and expression generated by the purple TF locus exceeds that generated by the gray TF locus. (Bottom) An eQTL variant disrupts the affinity of the gray TF, while the purple TF binds binds to the gene's *cis*-regulatory region and interacts multiplicatively with the region where the gray TF binds. The eQTL effect should remain constant at all purple TF levels.
(TIF)

**S2 Fig. Presence of lead eQTLs in filtered eVariant-eGene pairs. (Top)** The presence of lead eQTLs that are found in the filtered eVariant-eGene set are plotted by tissue. The inset displays the total percent of lead tissue eQTLs in the filtered set. **(Bottom)** The presence of eGenes that are found in the filtered eVariant-eGene set are plotted by tissue. The inset displays the percent of all eGenes that are in the filtered set.
(TIF)

**S3 Fig. Variant-gene associations of potentially causal eQTL variants.** Caviar fine-mapped variants were overlapped with TF ChIPseq and motif variants. The number of resulting fine-mapped variants per gene (top) and genes per variant (bottom) are displayed.
(TIF)

**S4 Fig. Clustering of GTEx tissues based on gene expression.** We clustered tissues based on median TPM across all genes using Euclidean distances and Ward.D clustering, cut the resulting tree to generate twenty clusters (green boxes). The tissue with the largest sample size was selected from each cluster.
(TIF)

**S5 Fig. Within-tissue TF-eQTLs.** 5% FDR TF-eQTLs per tissue, colored by whether or not top TF-eQTL variant was significantly associated with gene expression in that tissue. Only TF-eQTL variants with a significant eQTL were retained for further analysis. (Bottom) Tissues plotted by number of TF-eQTLs vs. tissue sample size. Outlier tissues are labeled.
(TIF)

**S6 Fig. Cell type composition variability of GTEx tissues.** Cell type enrichments were calculated *in silico* using XCell, and the mean/variance of each cell type in each tissue was calculated. Both plots show cell type variance vs. mean per tissue (dot color). aDC and iDC estimates frequently had large variance (left), thus they were removed (right). Four tissues remained with large cell type variance: fibroblasts, colon, stomach, and blood.
(TIF)

**S7 Fig. Histogram of 5% FDR TF-eQTLs across tissues.** The majority of TF-eQTLs were seen in one tissue only, though 4318/40065 (10.8%) of TF-eQTLs were observed in more than one tissue.
(TIF)

**S8 Fig. Cross-tissue TF-eQTLs per TF.** Number of cross-tissue expression-based TF-eQTLs are plotted versus number of cross-tissue protein-based TF-eQTLs. Dots are colored by Spearman correlation of cross-tissue median TF protein and expression levels.
(TIF)

**S9 Fig. TF protein levels.** Log10 of median relative protein abundance for all measured proteins, subset by whether the protein is a tested TF or not.
(TIF)

**S10 Fig. Annotation overlap of top TF-eQTL variants.** Enrichment of top TF-eQTL variant in each dataset for various genomic annotations from Ensembl Variant Effect Predictor.
(TIF)

**S11 Fig. TFBS overlap of top TF-eQTL variants.** Enrichment of top TF-eQTL variant in each dataset for TF overlap, as defined by ENCODE TF ChIPseq peaks, HOCOMOCO predicted motifs, or both annotations together. Fisher's exact test p-values are plotted.
(TIF)

**S12 Fig. Tested variants per gene.** Number of tested variants per gene, separated by whether the gene had a TF-eQTL (color) in the given dataset (x-axis). Significant TF-eQTL genes tended to have more tested variants per gene than unsignificant genes.
(TIF)

**S13 Fig. TF binding overlap enrichment.** Overlap enrichment for each TF-eQTL dataset for ChIPseq overlap, motif overlap, and both overlap.
(TIF)

**S14 Fig. Expected ASB overlap.** Number of expected TF-eQTL genes with ASB calculated by number of TF-eQTL gene variants accessible for a TF times percent of all accessible variants with ASB for that TF.
(TIF)

**S15 Fig. ASE in HEK293T cells for dual-evidence TF-eQTL genes.** Read counts are plotted for reference (orange) and alternative (blue) coding variants in IRF1-eQTL genes. Promoter-knockdown samples are in light shade and control samples are in dark shade. IRF1-eQTL genes may be heterozygous for a top IRF1-eQTL variant (top) or not (bottom). Fisher's exact test p value for data combined across timepoints is displayed at the top of each plot, while significance for individual timepoints is denoted by symbols: + $p<0.10$, * $p<0.05$, ** $p<0.01$. We see that three genes with heterozygous IRF1-eQTL variants show imbalanced ASE across conditions (*ERI1*, *MYOM2*, *RP5-1092A3.4*), while no genes without a heterozygous IRF1-eQTL variant.
(TIF)

**S16 Fig. P values of IRF1 knockdown effects on ASE.** Fisher's exact tests were run on 2x2 tables of allelic read counts in IRF1 knockdown and control experiments for all adequately covered genes.
(TIF)

**S17 Fig. LocusZoom plots of GWAS and eQTL.** A) P values are plotted for four blood cell GWAS traits that colocalize with an *APBB1IP* eQTL in any tissue. B) P values are plotted for *APBB1IP* eQTL in three tissues with an IKZF1-eQTL signal. Top IKZF1-eQTL variants are labeled, as well as a variant that overlaps an IKZF1 motif and ChIP-seq peak (overlap).
(TIF)

**S18 Fig. IKZF1-eQTL interaction signals.** Individuals are plotted by *APBB1IP* expression vs. IKZF1 TF expression in three tissues. Trend lines show linear regression lines per genotype. All three tissues had a significant IKZF1-eQTL.
(TIF)

**S19 Fig. TF binding information of *rs1335540*.** rs1335540 (red line) overlaps multiple TF ChIPseq peaks and two TF motifs. IKZF1 is the only TF which the SNP overlaps both of. The IKZF1 motif logo is displayed at the top of the figure, as calculated in HOCOMOCO v11 [50].
(TIF)

**S1 Table. GTEx Tissues.** Information on available expression and protein samples for GTEx tissues.
(XLSX)

**S2 Table. Dual Evidence TF-eQTLs.** TF-eQTL genes with two lines of supporting evidence (2 + tissues or cross + within tissue). Each row represents a TF-eQTL in a single dataset.
(XLSX)

**S3 Table. GxE Overlap.** Overlap of dual evidence TF-eQTL genes with gene-by-environment interacting genes from Findley et al., 2021.
(XLSX)

**S4 Table. GWAS Coloc TF-eQTLs.** eQTL genes in tissues that both colocalize with a GWAS trait and have a dual evidence TF-eQTL in that tissue, filtered for $r^2 > 0.4$ between the lead TF-eQTL variant and lead colocalizing variant.
(XLSX)

**S5 Table. Colocalizations between *APBB1IP* eQTLs and GWAS traits.**
(XLSX)

# Acknowledgments

We thank all members of the Lappalainen laboratory for their valuable discussions, and especially Stephane Castel, Margot Brandt, and Paul Hoffman for their data contributions. We thank the GTEx Consortium for their data, analyses, and expertise, including Andrew Brown and Farhad Homozdiari for their fine-mapping analyses, Lihua Jiang and Michael Snyder for access to GTEx protein data, and Hae Kyung Im, Alvaro Barbeira, and Rodrigo Bonazzola for the GWAS harmonization and colocalization efforts. We also thank Ivan Kulakovskiy and Vsevolod Makeev and their research teams for access to the ADASTRA allele-specific binding dataset.

# Author Contributions

**Conceptualization:** Elise D. Flynn, Pejman Mohammadi, Tuuli Lappalainen.

**Data curation:** Elise D. Flynn, Athena L. Tsu, Silva Kasela, Sarah Kim-Hellmuth, Francois Aguet.

**Formal analysis:** Elise D. Flynn, Athena L. Tsu.

**Funding acquisition:** Elise D. Flynn, Tuuli Lappalainen.

**Investigation:** Elise D. Flynn, Athena L. Tsu, Sarah Kim-Hellmuth.

**Methodology:** Elise D. Flynn, Pejman Mohammadi, Tuuli Lappalainen.

**Resources:** Kristin G. Ardlie, Tuuli Lappalainen.

**Software:** Elise D. Flynn, Athena L. Tsu, Silva Kasela, Sarah Kim-Hellmuth.

**Supervision:** Harmen J. Bussemaker, Pejman Mohammadi, Tuuli Lappalainen.

**Validation:** Elise D. Flynn, Athena L. Tsu.

**Visualization:** Elise D. Flynn.

**Writing – original draft:** Elise D. Flynn, Athena L. Tsu, Tuuli Lappalainen.

**Writing – review & editing:** Silva Kasela, Sarah Kim-Hellmuth, Harmen J. Bussemaker, Pejman Mohammadi.

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
