## [Decision Letter · Decision Letter 0]

2 Sep 2021

Dear Dr Flynn,

Thank you very much for submitting your Research Article entitled 'Transcription factor regulation of eQTL activity across individuals and tissues' to PLOS Genetics.

The manuscript was fully evaluated at the editorial level and by independent peer reviewers. The reviewers appreciated the attention to an important problem, but raised some substantial concerns about the current manuscript. Based on the reviews, we will not be able to accept this version of the manuscript, but we would be willing to review a much-revised version. We cannot, of course, promise publication at that time.

If you decide to revise the manuscript for further consideration at PLOS Genetics, please aim to resubmit within the next 60 days, unless it will take extra time to address the concerns of the reviewers, in which case we would appreciate an expected resubmission date by email to plosgenetics@plos.org.

[LINK]

We are sorry that we cannot be more positive about your manuscript at this stage. Please do not hesitate to contact us if you have any concerns or questions.

Yours sincerely,

Francesca Luca

Guest Editor

PLOS Genetics

Scott Williams

Section Editor: Natural Variation

PLOS Genetics

Reviewer's Responses to Questions

**Comments to the Authors:**

Reviewer #1: Please see the uploaded attachment.

Reviewer #2: Extensive efforts have catalogued the regulatory architecture of gene expression variation across the human body, but less is known about the mechanisms associated with variation across other contexts. In the present study, Flynn, et al. propose the use of TF expression and eQTL effect size to identify putative upstream TF regulators of eQTL effects. This work represents a creative analysis approach by using the p-value from a [TF]*genotype interaction term in a regression framework and re-use of several publicly available data sets including GTEx, ENCODE, and a CRISPRi perturbation study. While the proposed method and its overall goals are of great interest, there are concerns with its current implementation.

Major points:

1. It is stated in the methods that a 20% FDR is used on the within-tissue TF-eQTL analysis because additional filtering was performed. Other than removing tissues with high cell type composition variability and the annotation overlap filtering, which left only 7% of 5% MAF SNPs in the list, what other filtering was performed? How many results survive a more conservative FDR threshold? Presenting and focusing on only the lenient 20% FDR results leaves one a bit skeptical.

2. The authors select eQTL signal variants using an intersection between genetic fine-mapped, ChIP-seq peak overlapping, and TF motif overlapping variants. How does this set of SNPs compare the lead SNP (or highest posterior probability fine-mapped SNP) at the signal? Is the lead or a very high LD SNP always selected? Also, why was a functional fine-mapping approach not used, as this seems like a natural implementation of the set operations that were performed?

3. There appear to be several circular analyses performed for enrichment testing of the discovered signals.

a. First, the approach used to test whether TF-eQTL interactions overlapped with TFBSs: given that the initial set of variants was filtered to only those that overlapped with one or more TF ChIP-seq peak and were present in a HOCOMOCO motif, one expects these variants to be enriched for those annotations. The authors should take this initial filtering approach into account when performing enrichment analyses.

b. Second, a similar circularity appears to happen for the GWAS colocalization enrichment analysis of the TF-eQTL signals. Again, initial filtering was performed for eQTL that overlap one or more TF ChIP-seq peak and were present in a HOCOMOCO motif. Thus, the entire eQTL list is not the appropriate null to compare for GWAS colocalization. Instead, the authors should look at the eQTL signals they tested after filtering and perhaps compare to see if there is a significant enrichment difference between the signals they report as significant versus the signals they tested but did not report as significant.

Minor points:

1. What is the overall similarity between the motifs of TFs tested, and what was the reasoning for using collapsing PWMs and using regex matching for motif analysis versus a more traditional PWM scanning approach?

2. Also related to the PWMs, in the methods section, it’s not clear what “trimming any less confident bases (lowercase letters) from the ends of the sequence” really means. What are the actual threshold used to determine less confident bases and ends of the sequence?

3. There are two SFigure 10s referenced in the text and in the supplement.

**Have all data underlying the figures and results presented in the manuscript been provided?**

Reviewer #1: Yes

Reviewer #2: Yes

PLOS authors have the option to publish the peer review history of their article (what does this mean?). If published, this will include your full peer review and any attached files.

Reviewer #1: No

Reviewer #2: No

---

## [Decision Letter · Decision Letter 1]

6 Jan 2022

Dear Dr Flynn,

We are pleased to inform you that your manuscript entitled "Transcription factor regulation of eQTL activity across individuals and tissues" has been editorially accepted for publication in PLOS Genetics. Congratulations!

Yours sincerely,

Francesca Luca

Guest Editor

PLOS Genetics

Scott Williams

Section Editor: Natural Variation

PLOS Genetics

Comments from the reviewers (if applicable):

Reviewer's Responses to Questions

**Comments to the Authors:**

Reviewer #1: The authors addressed all of my concerns. This is a very nice manuscript.

Reviewer #2: The authors have addressed all the comments. Congratulations on the nice work.

**Have all data underlying the figures and results presented in the manuscript been provided?**

Reviewer #1: Yes

Reviewer #2: Yes

PLOS authors have the option to publish the peer review history of their article (what does this mean?). If published, this will include your full peer review and any attached files.

Reviewer #1: No

Reviewer #2: No

**Data Deposition**

http://datadryad.org/submit?journalID=pgenetics&manu=PGENETICS-D-21-00934R1

**Press Queries**

---

## [Editor Report · Acceptance letter]

27 Jan 2022

PGENETICS-D-21-00934R1 

Transcription factor regulation of eQTL activity across individuals and tissues 

Dear Dr Flynn, 

We are pleased to inform you that your manuscript entitled "Transcription factor regulation of eQTL activity across individuals and tissues" has been formally accepted for publication in PLOS Genetics! Your manuscript is now with our production department and you will be notified of the publication date in due course.

With kind regards,

Anita Estes

PLOS Genetics

On behalf of:
